# Characterization of Pathway-Specific Regulator NigR for High Yield Production of Nigericin in *Streptomyces malaysiensis* F913

**DOI:** 10.3390/antibiotics11070938

**Published:** 2022-07-13

**Authors:** Junhong Wei, Mengting Ma, Senwen Guo, Yaobo Xu, Jie Xie, Guoqing Pan, Zeyang Zhou

**Affiliations:** 1State Key Laboratory of Silkworm Genome Biology, Chongqing Key Laboratory of Microsporidia Infection and Control, Southwest University, Chongqing 400715, China; weijunhong@swu.edu.cn (J.W.); gsw0715@email.swu.edu.cn (S.G.); xiejie_bio@swu.edu.cn (J.X.); 2College of Sericulture, Textile and Biomass Sciences, Southwest University, Chongqing 400715, China; 2008301043@st.gxu.edu.cn; 3School of Life Sciences, Southwest University, Chongqing 400715, China; xuyaobo@swu.edu.cn; 4College of Life Sciences, Chongqing Normal University, Chongqing 401331, China

**Keywords:** nigericin, *Streptomyces malaysiensis*, NigR, pathway-specific regulator

## Abstract

Nigericin is a polyether antibiotic with potent antibacterial, antifungal, antimalarial and anticancer activity. NigR, the only regulator in the nigericin biosynthetic gene cluster in *Streptomyces malaysiensis* F913, was identified as a SARP family regulator. Disruption of *nigR* abolished nigericin biosynthesis, while complementation of *nigR* restored nigericin production, suggesting that NigR is an essential positive regulator for nigericin biosynthesis. Overexpression of *nigR* in *Streptomyces malaysiensis* led to significant increase in nigericin production compared to the wild-type strain. Nigericin production in the overexpression strain was found to reach 0.56 g/L, which may be the highest nigericin titer reported to date. Transcriptional analysis suggested that *nigR* is required for the transcription of structural genes in the *nig* gene cluster; quantitative RT-PCR analysis revealed that the expression of structural genes was upregulated in the *nigR* overexpression strain. Our study suggested that NigR acts in a positive manner to modulate nigericin production by activating transcription of structural genes and provides an effective strategy for scaling up nigericin production.

## 1. Introduction

*Streptomyces* are well known for their extraordinary capacity to produce bioactive secondary metabolites in industrial, agricultural and medical applications. Secondary metabolism in *Streptomyces* is regulated in a complex fashion in response to environmental manipulations. The regulation of secondary metabolite biosynthesis is governed by various regulators, including cluster-situated regulators (CSRs), pleiotropic regulators and global regulators [1]. Global regulators and pleiotropic regulators are scattered throughout the chromosome, affecting the production of multiple antibiotics and/or morphological development [2]. CSRs usually located within the antibiotic biosynthetic clusters and exert their regulatory functions via modulating the transcription of biosynthetic genes or other regulatory genes. Among these CSRs, *Streptomyces* antibiotic regulatory protein (SARP) family transcriptional regulators constitute the most frequently occurring transcriptional regulators, which serve mainly as activators in antibiotic biosynthesis [3]. ActII-ORF4, the founding member of this family, is an activator of actinorhodin production in model strain *Streptomyces coelicolor* [4]. Binding of ActII-ORF4 to upstream of *actIV-orfA* operon and *actIV-orf1* operon activates their transcription [5]; the expressed structural genes then initiate the biosynthesis of actinorhodin. As with ActII-ORF4, RedD activated the production of undecylprodiginines in *S. coelicolor* [6]. SARP family regulators also function in many streptomycetes, such as SanG in *Streptomyces ansochromgenes* (nikkomycin cluster), CcaR in *Streptomyces clavuligerus* (cephamycin-clavulanic acid supercluster) and VlmI in *Streptomyces viridifaciens* (valanimycin cluster) [7,8,9]. Apart from directly activating the transcription of key structural genes, many other SARP family regulators function in different ways. Some pathways contain more than one SARP and form complex cascade regulation patterns. Representative members, include the PolY of polyoxin biosynthesis in *Streptomyces cacaoi* subsp. *asoensis* [10], which controls the biosynthesis of polyoxin indirectly through modulating the expression of *polR*, the activator of key structural genes such as *polC* and *polB* [11]. Though most SARP family regulators function as cluster-suited regulators which activate production of the cognate antibiotic biosynthesis cluster, examples of other roles have also been reported. The SARP regulator AfsR serves as a global regulator and appears to be present in all streptomycetes [12]. 

Polyether antibiotics are a unique class of ionophores with metal ion-binding ability and lipophilic characteristics [13]. One major worldwide commercial use of polyether antibiotics is to control coccidiosis in animal husbandry [14,15]. The first discovered carboxyl polyether antibiotic was nigericin (Figure 1a). It has the ability to selectively transport ions across cell membranes and is widely used as a research tool to disrupt intracellular H^+^ and K^+^ concentration [16,17]. Nigericin is reported to be a potent agent against a variety of multidrug-resistant strains of pathogens, including *Paecilomyces variotii* and *Candida albicans* [18]. In vivo and in vitro assays have also proved that nigericin exerts strong selective antitumor activities [19,20]. The entire gene cluster for nigericin biosynthesis was firstly cloned from *Streptomyces* sp. DSM4137; the biosynthetic pathway of nigericin was proposed based on analyses of intermediates accumulated in disruption mutants of several structural genes [21]. However, the regulation mechanism of nigericin biosynthesis is still unknown, rendering improvement of the production of nigericin production by genetic engineering problematic. *Streptomyces malaysiensis* F913 is a soil-borne actinomycete with high algicidal activity [22]. Whole genome scanning suggested that *S. malaysiensis* F913 has a nigericin biosynthetic gene cluster (*n**ig* gene cluster) which showed high synteny with the reported homologous gene cluster in *Streptomyces* sp. DSM413. In this paper, we determined the role of the only regulatory gene, *nigR*, in the nigericin biosynthesis gene cluster in *S. malaysiensis* F913. The results suggest that NigR positively regulated nigericin biosynthesis by activating the transcription of structural genes. Overexpression of *nigR* significantly improved nigericin production. 

## 2. Results

### 2.1. NigR Encodes a Putative SARP Family Transcriptional Regulator

Assembly of the sequencing data of the *S. malaysiensis* F913 genome yielded 25 contigs, assembled into 6 scaffolds (GenBank accession number: LJIW00000000). The fine genome map of *S. malaysiensis* F913 was 11,608,319 bp in length without any plasmid, the chromosome encoded 9497 predicated genes, and the total length of the genes was 10,011,963 bp. The *nig* gene cluster was identified by searching the genome sequence of *S. malaysiensis* F913. As the nigericin biosynthesis gene cluster in *S. malaysiensis* F913 showed high synteny with the reported cluster in *Streptomyces* sp. DSM413, we used the same gene name as proposed for the original discovered cluster (Appendix A). Among the 20 genes in the predicated *nig* gene cluster in *Streptomyces malaysiensis* F913, *nigR* was the only transcriptional regulator (Figure 1b). Sequence analysis of NigR was performed with BLAST (http://www.blast.ncbi.nlm.nih.gov/Blast.cgi (accessed on 25 May 2022) and SMART (http://smart.embl-heidelberg.de (accessed on 25 May 2022). The results suggested that NigR is a SARP family regulator, with an N-terminal trans-Reg-C domain (amino acids 19–96) (Pfam no. PF00486) which may play a role in DNA binding, and a bacterial transcriptional activator domain (amino acids 103–248) (Pfam no. PF03704) located in the C-terminal (Figure 2a). NigR shows 63% identity to ChlF2 (AAZ77687.1), 58% identity to MonR1 (ANZ52472.1), 57% identity to NanR2 (AAP42854.1) and 50% identity to NanR1 (AAP42853.1) (Figure 2b). ChlF2 is a cluster-situated SARP family regulator which is essential for chlorothricin biosynthesis in *Streptomyces antibioticus* DSM 40725 [23]. MonR1 positively regulated monensin biosynthesis in *Streptomyces cinnamonensis* ST021 [24]. Both NanR1 and NanR2 are transcriptional activators of polyketide genes in the nanchangmycin biosynthesis gene cluster in *Streptomyces nanchangensis* NS3226 [25]. A rare TTA codon was located in codon 22 in NigR, suggesting that the translation of *nigR* may rely on the *bldA* gene [26].

### 2.2. Disruption of NigR Abolished Nigericin Production

To determine the role of *nigR* in nigericin biosynthesis, a *nigR* disruption mutant was constructed via double-crossover recombination. The resulting mutant strain (DM01R) was tested for nigericin production. Unlike the *S. malaysiensis* F913 wild-type strain (WT), nigericin production was abolished in the *nigR* disruption mutant (Figure 3). To prove that the increase in nigericin production was attributable to the effect *nigR* disruption, complementation strain DM01Rc was constructed by integrating a copy of *nigR* with native promoters into the chromosome of DM01R by pSET152. The results show that nigericin production was restored in the complementary strain, and further confirmed that NigR is a key activator of nigericin biosynthesis in *S. malaysiensis* F913 (Figure 3). The DM01R strain showed patterns of growth and morphology identical to those of the wild-type strain on MS or Gause’s medium, indicating that *nigR* has no significant effect on the growth and differentiation of *S. malaysiensis* F913. 

### 2.3. Gene Expression Analysis in the Wild-Type and Strain DM01R

In order to determine the regulation mechanism and potential targets of NigR, reverse transcription PCR (RT-PCR) tests were performed to detect the transcription of the gene in the *nig* gene cluster. RT-PCR products of the all genes in the *nig* gene cluster were detected in the *S. malaysiensis* F913 strain after 27 cycles of PCR, while no band could be observed in the DM01R strain, indicating that these genes are probably regulated by NigR (Figure 4). A primer pair designed to amplify the cDNA of *hrdB*, encoding an essential sigma factor in *S. malaysiensis* (accession no. PNG92438.1), was used as internal control. The results suggest that NigR is essential for the expression of structural genes; the promoters of structural genes are potential targets of NigR. A consensus sequence of “CGWWWCCG” was identified in the promoter of *orf9*, intergenic region of *nigD-C* and *nigAI-nigE* (Appendix A).

### 2.4. Overexpression of NigR Significantly Enhanced Production of Nigericin

In general, overexpression of a pathway-specific positive regulator will lead to increased production of the corresponding antibiotics. To examine the effects of *nigR* overexpression in *S. malaysiensis* F913, we constructed *nigR* overexpression vectors pSET152::P*_hrdB_*R and pSET152::P*_kasO_*R. pSET152, pSET152::P*_hrdB_*R and pSET152::P*_kasO_*R were introduced into *S. malaysiensis* F913 to generate F913-pSET152, F913-hrdBR and F913-kasOR, respectively. Nigericin production was increased in both F913-hrdBR and F913-kasOR strains compared to F913 and F913-pSET152 controls. Compared to the F913 strain, nigericin production rose by 68% in F913-hrdBR and 54% in F913-kasOR at day five (Figure 5a). The biomass of *S. malaysiensis* F913, F913-pSET152, F913-hrdBR and F913-kasOR were determined to exclude the effect of growth on antibiotic production. The results showed that F913, F913-pSET152, F913-hrdBR and F913-kasOR showed comparable growth rates and final dry weights (Figure 5b), indicating that the increase in nigericin production could be attributed to the effects of *nigR* overexpression.

### 2.5. Gene Expression Analysis in the Wild-Type and Overexpression Strains

To determine the effects of *nigR* overexpression on nigericin biosynthetic genes, qRT-PCR analysis was performed to assess the transcription of genes involved in the *nig* gene cluster. Transcriptional analyses showed that the expression levels of *nigR*, *nigAVII*, *nigD*, *nigCII*, *nigAI* and *nigE* increased dramatically in *nigR* overexpression strains F913-hrdBR and F913-kasOR, compared with the unmodified wild-type strain (Figure 6). In addition, the expression of *nigR* also increased significantly in the F913-hrdBR and F913-kasOR strains. These results further proved that the enhanced production of nigericin in the overexpression strains can be attributed to increased expression of *nigR* and subsequent activation of nigericin biosynthetic genes.

## 3. Discussion

Nigericin has a broad spectrum of biological activities. Recent research has focused on its anticancer activities [19,27]. Previous studies have analyzed the detailed mechanism of nigericin biosynthesis; however, little is known about the regulation of nigericin biosynthesis. In this study, a SARP family activator, NigR—the only regulator in the nigericin biosynthetic gene cluster—was identified. Sequence analysis indicated that NigR has an N-terminal DNA-binding domain and an adjacent transcriptional activation domain. The presence of TTA codons in coding regions suggests that the translation of NigR may be regulated by *bldA*. RT-PCR analysis revealed that NigR is essential for the expression of structural genes, suggesting that these genes are the potential targets of NigR. Unfortunately, all attempts to overexpress and purify recombinant NigR proteins failed, and we were unable to perform gel mobility shift assays to identify the direct target genes of NigR. In a future study, we will try GUS (β-glucuronidase) assays to detect the interaction of NigR with promoters of target genes [28]. 

Combinational mutation and culture condition optimization have been carried out to improve nigericin production; however, titer improvement of nigericin production by overexpressing regulators has not been reported [29,30]. After decades of extensive research about the regulation of the production of secondary metabolites in actinomycetes, much is known particularly about *Streptomyces* species. Secondary metabolite titers can be influenced by global regulatory networks and pathway-specific regulation, which makes these regulators perfect targets for the rational engineering of overproduction strains. Overexpression of positive regulatory genes is a well-established strategy for strain improvement. By multiplying copy numbers of *sanG*, nikkomycin production was increased significantly [31]. In *Streptomyces fradiae*, insertion of a second copy of the positive regulator *tylR* under control of the strong constitutive *ermE** promoter caused a 50% improvement in tylosin production [26]. The *ermE** promoter from *Saccharopolyspora erythraeus* is the most widely used strong constitutive promoter; other optional tools, such as SF14p, are also available [32]. In recent years, strong and constitutive promoters such as *kasO** promoter and *hrdB* promoter, have been developed, and have shown better performance in some *Streptomyces* species [33,34]. In this work we used *kasO** promoter and *hrdB* promoter as *cis* regulatory elements to drive *nigR* expression. The results suggest that both promoters can upregulate nigericin expression and the F913-hrdBR strain showed higher nigericin production. The mean production of nigericin in the F913-hrdBR strain was 0.56 g/L. To our knowledge, this is the highest nigericin titer reported to date, representing a new step for scaling-up the production of nigericin. Promoter strength varied in different *Streptomyces* strains, implying that more promoters need to be screened to choose the most suitable one for strain improvement in *S. malaysiensis* F913. Secondary metabolites are synthesized from precursors through multistep biosynthetic pathways with the cells. Medium and fermentation process optimization are also required to further increase the production of nigericin in engineered overexpression strains. 

## 4. Materials and Methods

### 4.1. Strains, Plasmids, and Growth Conditions

The bacterial strains and plasmids used in this study are listed in Appendix A. *Streptomyces malaysiensis* F913, a natural nigericin producer, was isolated from the soil of Southwest University; the accession number of its genome is LJIW00000000. *Streptomyces malaysiensis* F913 and their derivatives were grown on MS agar (2% mannitol, 2% soybean powder) at 28 °C. Liquid Gause’s medium (0.001% FeSO_4_·7H_2_O, 0.05% NaCl, 0.05% K_2_HPO_4_, 0.05% MgSO_4_·7H_2_O, 0.1% KNO_3_, 2% soluble starch) was used for nigericin production. All antibiotics were purchased from Sangon Biotech (Shanghai) Co., Ltd., Shanghai, China. 

### 4.2. Construction of NigR Mutants

To construct *nigR* disruption vectors, a 2329 bps PCR product from primer NigRdm-upF/R (Appendix A) was amplified and named as the upstream arm, which contained partial N-terminal *nigR* and adjacent flanking sequences. A 2585 bps PCR product from primer NigRdm-dnF/R (Appendix A) was amplified and named as the downstream arm, which contained partial C-terminal *nigR* and adjacent flanking sequences. The upstream arm was digested with *Hin*dIII and *Xba*I; the downstream arm was digested with *Xba*I and *B**am*HI. A kanamycin-resistance cassette was amplified from pUC119::*neo* by PCR with primer kanF/R, and digested with *Xba*I. Three resulting DNA fragments were inserted into the *Hin*dIII and *B**am*HI digested pSET152 to generate pSET152sv::*nigR*::*neoR* in which a 625 bp fragment of *nigR* was replaced by the kanamycin-resistance gene. As the ΦC31 integrase in pSET152 contained two *Hin*dIII sites and was digested by *Hin*dIII, the integrase coding region in pSET152sv::*nigR*::*neoR* was incomplete and this vector became a suicide vector. After restriction digestion analysis and PCR verification, pSET152sv::*nigR*::*neoR* was transformed into *E. coli* ET12567/pUZ8002 and conjugally transferred to *S. malaysiensis* F913 by intergeneric conjugation. After growing for 5 days at 28 °C, the colonies were replicated on MS agar plates containing kanamycin. The double cross-over exconjugants were apramycin sensitive and kanamycin resistant and these strains were then verified by PCR.

### 4.3. Complementation and Overexpression of NigR 

The coding region of *nigR* with its native promoter was amplified from the genomic DNA of *S. malaysiensis* F913 with primer pair nigRNP-F/R to generate nigRNP (Appendix A). NigRNP was digested with *Xba*I/*Eco*RI, then ligated with *Xba*I/*Eco*RI digested pSET152 to generate complementary vector pSET152::nigRc. The pSET152::nigRc was introduced into the DM01R strain to generate the DM01Rc strain. For construction of *nigR* overexpression vectors, the coding region of *nigR* was amplified from the genomic DNA of *S. malaysiensis* F913 with primer pair nigR-CDSF/R (Appendix A) and digested with *N**de*I/*E**co*RI to generate DNA fragment NigR-CDS. *hrdB* promoter was amplified from the genomic DNA of *S. malaysiensis* F913 with primer pair PhrdBF/R (Appendix A) and digested with *Nde*I/*Xba*I to generate DNA fragment PhradB. NigR-CDS and PhrdB were ligated into *Xba*I/*Eco*RI digested pSET152 to generate pSET152::P*_hrdB_*R. To construct *kasO** promoter, oligonucleotides PkasOF and PkasOR (Appendix A) were mixed in a tube and heated to 95 °C for 5 min; the tube together with its content was then incubated into a beaker containing 200 mL water at 95 °C and cooled down slowly to room temperature to anneal these two oligonucleotides to PkasO, a double-strand DNA fragment with sticky ends on both sides. NigR-CDS and PkasO were ligated into *Xba*I/*Eco*RI digested pSET152 to generate pSET152::P*_kasO_*R. After restriction digestion analysis and PCR verification, pSET152::P*_hrdB_*R and pSET152::P*_kasO_*R were transformed into *E. coli* ET12567/pUZ8002 and conjugally transferred to *S. malaysiensis* F913 to generate F913-hrdBR and F913-kasOR, respectively. 

### 4.4. Production and Analysis of Nigericin

For nigericin production, spore suspensions were inoculated into liquid Gause’s medium and incubated for 168 h at 28 °C before sampling. For analysis of nigericin, culture supernatants were extracted twice with an equal volume of ethyl acetate, and mycelia were extracted with 50 mL of methanol. Combined extracts were dried by removing the solvent under reduced pressure at 50 °C. The dried pellet was then dissolved in 1 mL methanol. High-performance liquid chromatography (HPLC) analysis was performed on an Agilent 1260 HPLC system and ZORBAX SB-C18 column (5 µm, 4.6 × 250 mm) (Agilent Technologies, Inc., Santa Clara, CA, USA). HPLC conditions were as follows: a linear gradient of MeOH/H_2_O from 80:20 to 100:0 over 15 min, 100% MeOH for 5.5 min and 20% MeOH + 80% H_2_O for 6 min, flow rate = 1 mL/min, UV = 305 nm. For determination of cell dry weight, mycelium from 20 mL cell cultures were collected and dried at 65 °C for 16 h to constant weight. All experiments are carried out in triplicate. Significance was analyzed by means of a Student’s *t*-test (GraphPad Prism 6).

### 4.5. RNA Extraction, RT-PCR and qRT-PCR

The primers used in RT-PCR and qRT-PCR are listed in Appendix A. RNA extraction, genomic DNA removal, cDNA Synthesis and reverse-transcription PCR (RT-PCR) were the same as described [11]. For the RT-PCR experiment, total RNAs were extracted from the *S. malaysiensis* F913 and DM01R strains after inoculation for 96 h and used as templates for analysis of gene expression. For qRT-PCR analysis, total RNAs were isolated from mycelium of F913, F913-hrdBR and F913-kasOR strains grown in Gause’s medium at various fermentation times (48, 72, 96, 120 and 144 h). The qRT-PCR was performed by a LightCycle 96 instrument (Roche, Zurich, Switzerland). All experiments were carried out in triplicate. The relative expression levels of all samples were calculated using the 2^−ΔΔCT^ method [35]. All qRT-PCR values were normalized to the endogenous control *hrdB* (a housekeeping sigma factor in *Streptomyces*). Significance tests were calculated by means of a Student’s *t*-test (GraphPad Prism 6).

### 4.6. Genome Sequencing, Assembly and Annotation

The genomic DNA of *S. malaysiensis* F913 was extracted, followed by RNase treatment. The draft genome sequence of *S. malaysiensis* F913 was achieved using Illumina Hiseq2000 sequencing system at NovoGene Bioinformatics Institute in Beijing, China. The paired-end reads generated by the Illumina sequencer were assembled by using SOAPdenovo 1.05 [36]. Functional gene annotation was based on Rapid Annotation using Subsystem Technology (RAST) [37]. The putative secondary metabolites’ biosynthetic gene clusters were further specifically identified and categorized using antibiotics and a Secondary Metabolite Analysis Shell [38].

### 4.7. Nucleotide Sequence Accession Number

The nucleotide sequences of *nig* gene clusters determined in this study have been submitted to the GenBank database under accession number ON664936.

## 5. Conclusions

NigR, the only regulator in the nigericin biosynthetic gene cluster in *S. malaysiensis* F913, was identified as a SARP family regulator in this study. Disruption of *nigR* resulted in loss of nigericin production, while complementation of *nigR* restored nigericin production. The targets of NgrR might be structural genes in the *nig* gene cluster, given the knockout of ngrR abolishes the transcription of all biosynthesis genes. Nigericin overproducing strains were constructed through overexpression of *nigR*. The maximum production of nigericin was achieved in the F913-hrdBR strain at about 0.56 g/L, which was about 1.5-fold higher than *S. malaysiensis* F913. These results extended our understanding about the regulatory mechanism of nigericin biosynthesis, and have also provided a promising strategy for scaling up the production of nigericin.

## Figures and Tables

**Figure 1 antibiotics-11-00938-f001:**
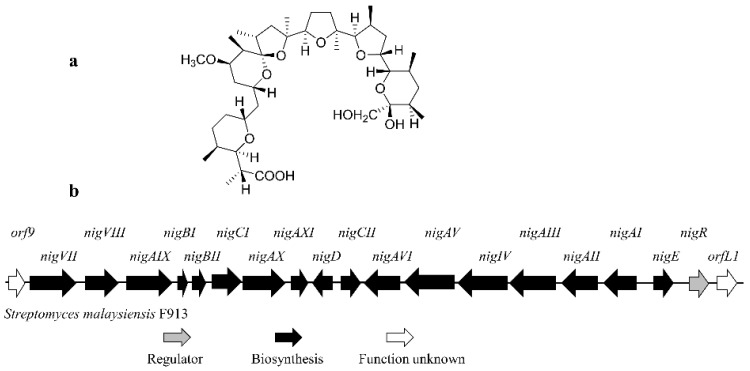
Structure and biosynthetic gene cluster of nigericin. (**a**) Structure of nigericin; (**b**) Biosynthetic gene cluster of nigericin in *Streptomyces malaysiensis* F913. The gene cluster spans 97.9 kb fragment and contains 20 ORFs. Genes are indicated by different arrowheads according to their proposed roles.

**Figure 2 antibiotics-11-00938-f002:**
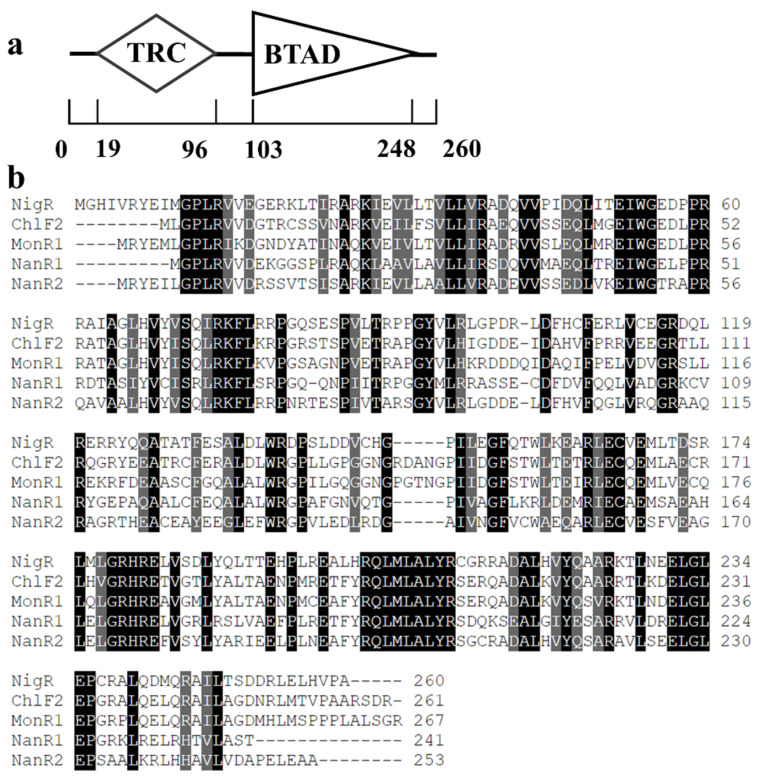
Identification of NigR in *S. malaysiensis*. (**a**) Predicted domain structure of NigR. TRC, trans-Reg-C domain; BTAD, bacterial transcriptional activator domain; (**b**) Amino acid alignment of NigR and its orthologs. Identical or similar residues in all sequences are highlighted in black and gray, respectively. Examples of orthologs include: ChlF2 for chlorothricin from *S. antibioticus*, MonR1 for monensin from *S. cinnamonensis*, NanR1 and NanR2 for nanchangmycin from *S. nanchangensis*.

**Figure 3 antibiotics-11-00938-f003:**
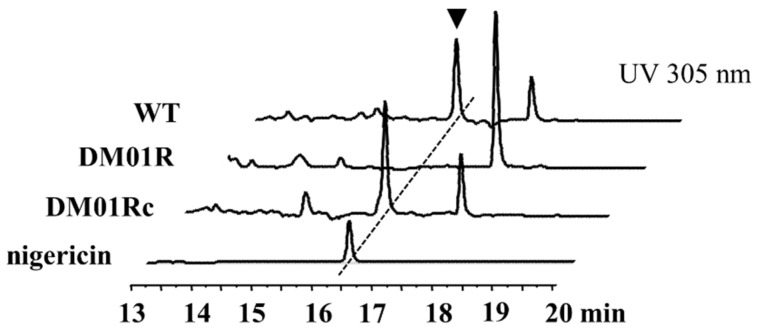
HPLC analysis of nigericin produced by *S. malaysiensis* F913 and its derivatives. WT, fermentation broth from *S. malaysiensis* F913; DM01R, fermentation broth from *S. malaysiensis nigR* disruption mutant; DM01Rc, fermentation broth from *S. malaysiensis nigR* complementation strain; nigericin: nigericin standard sample. The retention time for nigericin was about 16.7 min, indicated by triangle in WT.

**Figure 4 antibiotics-11-00938-f004:**
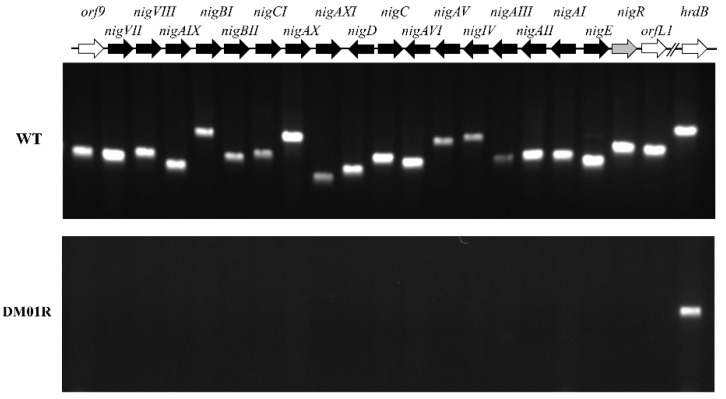
Transcriptional analysis of the *nig* gene cluster by RT-PCR. WT: the RT-PCR products of the RNAs extracted from the wild-type strain; DM01R: the RT-PCR products of the RNAs extracted from the *nigR* disruption mutant strain. Genes represented by arrows; an *hrdB* transcript was used as an internal control. Twenty-seven cycles of PCR were routinely employed.

**Figure 5 antibiotics-11-00938-f005:**
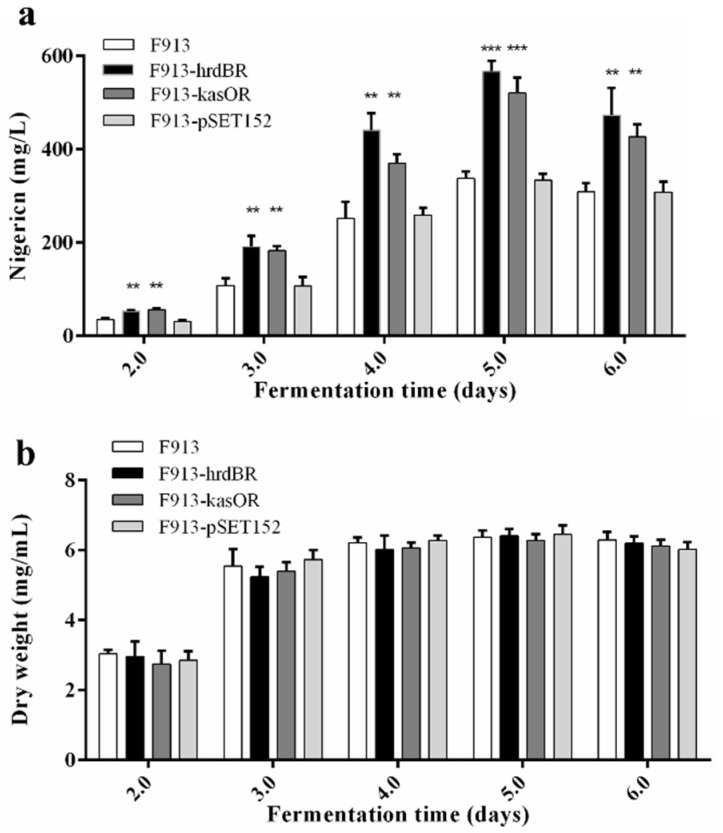
Effects of *nigR* overexpression on nigericin production. (**a**) Nigericin production; (**b**) Dry weight in liquid culture. Data are presented as the averages of three independent experiments conducted in triplicate. Error bars show standard deviations from three replicates. Student’s t -test is used to determine the *p*-values. Significant difference: ** *p* < 0.01, *** *p* < 0.001.

**Figure 6 antibiotics-11-00938-f006:**
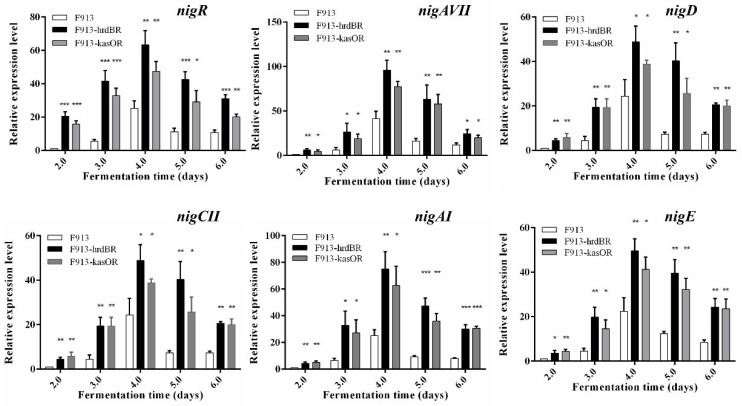
qRT-PCR analysis of selected genes in nigericin biosynthesis. The expression level of samples collected from the F913 strain at fermentation day 2 was assigned a value of 1; the expression levels of *nigR*, *nigAVII*, *nigD*, *nigCII*, *nigAI* and *nigE* in in all samples are presented relative to that of samples collected from the F913 strain at fermentation day 2. Error bars show standard deviations from three independent experiments. A Student’s *t*-test was used to determine the *p*-values. Significant difference: * *p* < 0.05, ** *p* < 0.01, *** *p* < 0.001.

## Data Availability

The datasets supporting the conclusions of this article are included within the article and the Appendix A.

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
