# Peer review of "Characterization of Pathway-Specific Regulator NigR for High Yield Production of Nigericin in Streptomyces malaysiensis F913"

_antibiotics, 2022, doi:10.3390/antibiotics11070938_

Round 1
Reviewer 1 Report
The authors characterize the role of NgrR, a transcriptional regulator of the SARP family, in the transcriptional control of the gene cluster directing the production of nigericin, a polyether antibiotic produced by Streptomyces malaysiensis.
The nig biosynthetic gene cluster was previously identified by Harvey et al (2007) and characterized for its functional roles in the synthesis of the compound, but has not yet been studied in the aspect of regulation.
In this paper, the authors studied the role of NgrR, the only transcriptional regulator encoded in the nig gene cluster, and revealed its involvement in the expression of the whole gene cluster. The evidence supports that NgrR is a positive transcriptional regulator.
Overall, experiments are well performed, and results fully support the role of NgrR as a positive transcriptional regulator. However, this reviewer feels that the article needs to be reconsidered with regard to the following points.
Line 70-
The authors refer to the availability of whole genome sequence information of S. malaysiensis in the Introduction. Is this published elsewhere? If so, please cite the reference. If genome sequencing is performed in this study, the details should be described in the section Results and Materials and Methods.
Line 84 & Fig. 1b
The authors named the nigericin biosynthetic gene cluster ‘ngr’, despite the original description of the gene cluster by Harvey et al. (2007) designate it to be ‘nig’ cluster. Since the gene arrangement within these clusters appears identical, this reviewer feels that the authors should use the same gene name as proposed by the original discoverer to avoid confusion. It would be useful for the authors to summarize the amino acid sequence identity between the two orthologs of each Nig protein.
Line 110
The HPLC chromatogram clearly shows the absence of nigercin in the culture broth of the ngrR mutant. On the other hand, the mutant appears to be overproducing the compound corresponding to the peak at 18 min. Is there any information regarding this phenomenon?
L130
The RT-PCR experiment clearly shows that the knockout of ngrR abolishes the transcription of all biosynthesis genes. This makes it likely that the NgrR protein regulates the transcription at the promoters that direct the expression of these genes. This reviewer feels that the authors can further assess this possibility by comparing the sequence of the promoters and identifying any consensus sequence.
L183
The authors comment that the recombinant protein preparation for NgrR was unsuccessful and therefore detailed characterization of its action could not be performed. Although this situation can be understood, this reviewer feels that the paper lacks information regarding the function of the regulator. Have the authors attempted to express a truncated form of NgrR containing the DNA binding domain?
Author Response
The authors characterize the role of NgrR, a transcriptional regulator of the SARP family, in the transcriptional control of the gene cluster directing the production of nigericin, a polyether antibiotic produced by Streptomyces malaysiensis.
The nig biosynthetic gene cluster was previously identified by Harvey et al (2007) and characterized for its functional roles in the synthesis of the compound, but has not yet been studied in the aspect of regulation.
In this paper, the authors studied the role of NgrR, the only transcriptional regulator encoded in the nig gene cluster, and revealed its involvement in the expression of the whole gene cluster. The evidence supports that NgrR is a positive transcriptional regulator.
Overall, experiments are well performed, and results fully support the role of NgrR as a positive transcriptional regulator. However, this reviewer feels that the article needs to be reconsidered with regard to the following points.
Line 70-
The authors refer to the availability of whole genome sequence information of S. malaysiensis in the Introduction. Is this published elsewhere? If so, please cite the reference. If genome sequencing is performed in this study, the details should be described in the section Results and Materials and Methods.
Response: Detailed genome sequence analysis and biosynthesis gene cluster predication are included in another manuscript. We herein briefly described whole genome sequence information in section “Results and Materials” and “Methods” according to suggestions of the reviewer.
Line 84 & Fig. 1b
The authors named the nigericin biosynthetic gene cluster ‘ngr’, despite the original description of the gene cluster by Harvey et al. (2007) designate it to be ‘nig’ cluster. Since the gene arrangement within these clusters appears identical, this reviewer feels that the authors should use the same gene name as proposed by the original discoverer to avoid confusion. It would be useful for the authors to summarize the amino acid sequence identity between the two orthologs of each Nig protein.
Response:We have changed it to “nig” as suggested. The deduced functions of genes in the nigericin biosynthetic gene cluster and amino acid sequence identity between the two orthologs of each Nig protein have been summarized in table S3.
Line 110
The HPLC chromatogram clearly shows the absence of nigercin in the culture broth of the ngrR mutant. On the other hand, the mutant appears to be overproducing the compound corresponding to the peak at 18 min. Is there any information regarding this phenomenon?
Response: Yes. We also noticed this phenomenon. Currently, we are working on the purification and identification of this peak. However, this compound is beyond the scope of this paper. Thus, we did not include it in this manuscript.
L130
The RT-PCR experiment clearly shows that the knockout of ngrR abolishes the transcription of all biosynthesis genes. This makes it likely that the NgrR protein regulates the transcription at the promoters that direct the expression of these genes. This reviewer feels that the authors can further assess this possibility by comparing the sequence of the promoters and identifying any consensus sequence.
Response: This is a very good suggestion. We compared the sequence of the promoters of target genes and identified a consensus sequence of “CGWWWCCG”. This information was included in the revised manuscript.
L183
The authors comment that the recombinant protein preparation for NgrR was unsuccessful and therefore detailed characterization of its action could not be performed. Although this situation can be understood, this reviewer feels that the paper lacks information regarding the function of the regulator. Have the authors attempted to express a truncated form of NgrR containing the DNA binding domain?
Response:We have tried to express a truncated form of NgrR containing the DNA binding domain. Unfortunately, expression of the DNA binding domain also failed.
Reviewer 2 Report
In this study, NgrR was identified as an important regulator for the biosynthesis of nigericin production in Streptomyces malaysiensis F913 through deletion and expression analyses. The work is well done, the manuscript is well written, the graphics are very descriptive and clear, and the results are interesting, they will probably be useful for biotechnological purposes. I only have one comment. Please specify the name of each gene in figure 1b instead of putting a letter in each arrow head.
Author Response
In this study, NgrR was identified as an important regulator for the biosynthesis of nigericin production in Streptomyces malaysiensis F913 through deletion and expression analyses. The work is well done, the manuscript is well written, the graphics are very descriptive and clear, and the results are interesting, they will probably be useful for biotechnological purposes. I only have one comment. Please specify the name of each gene in figure 1b instead of putting a letter in each arrow head.
Response: We thank the reviewer for the comments on our manuscript. We have made the changes as suggested. Gene names were changed accordingly.

Reviewer 3 Report
In the manuscript by Wei et al., entitled: ‘Characterization of pathway-specific regulator NgrR for high yield production of nigericin in Streptomyces malaysiensis F913’, the authors determine the role of the only regulatory gene ngrR in the nigericin biosynthesis gene cluster in S. malaysiensis F913. Although the manuscript is of interest, there are some minor points to revise before possible publication on Antibiotics journal.
-The authors evaluated the increasing of production of nigericin without indicate the value of the yield. Please add this information also in the abstract and clarify this point;
-Figure 3. Please modify the Figure adding the wavelength used on y axis and the overlay B on the chromatogram and as secondary axis;
-Figures 5 and 6: which statistical tested used for the analysis? Please add information in Figure legend;
-The authors should add a ‘conclusions’ section in order to further improve the quality of the manuscript.
Author Response
In the manuscript by Wei et al., entitled: ‘Characterization of pathway-specific regulator NgrR for high yield production of nigericin in Streptomyces malaysiensis F913’, the authors determine the role of the only regulatory gene ngrR in the nigericin biosynthesis gene cluster in S. malaysiensis F913. Although the manuscript is of interest, there are some minor points to revise before possible publication on Antibiotics journal.
-The authors evaluated the increasing of production of nigericin without indicate the value of the yield. Please add this information also in the abstract and clarify this point;
Response: The abstract has been revised as suggested.
-Figure 3. Please modify the Figure adding the wavelength used on y axis and the overlay B on the chromatogram and as secondary axis;
Response: The figure 3 has been revised as suggested
-Figures 5 and 6: which statistical tested used for the analysis? Please add information in Figure legend;
Response: Student’s t-test was used for the analysis. This information was added in figure legends of the revised manuscript.
-The authors should add a ‘conclusions’ section in order to further improve the quality of the manuscript.
Response: This is a good suggestion. A new section of Conclusion was added as suggested.

Reviewer 4 Report
Overview
The manuscript submitted by Wei and coworkers details the investigation of a putative regulator (NgrR) of the nigericin biosynthetic gene cluster present in Streptomyces malaysiensis F913. The authors predict that the NgrR protein belongs to the SARP family of regulators using sequence analysis, and show that disruption of the ngrR gene abolishes nigericin production and transcription of nigericin biosynthetic genes. Overexpression of NgrR using kasO and hrdB promoters resulted in an increase in nigericin production relative to the wild-type strain.
General Comments
Engineering promoter systems in Streptomyces to boost the production of natural products has proven to be a successful strategy in many instances, and so the characterization of regulatory genes associated with BGCs of interest such as the nigericin BGC is clearly of value to the field. It would've been interesting to see if introducing multiple copies of ngrR into the strain would've further boosted the production (as has been shown in other examples), but perhaps this is beyond the scope of this study.
Specific Comments
1. From the chromatograms shown in Fig. 3, there's another peak (RT 18 mins) which appears to increase in intensity when ngrR is disrupted. Do the authors know which metabolite this peak corresponds to? Is this peak also associated with the nigericin cluster? The authors should comment on this (perhaps as a note in the figure legend if it's not significant).
2. In the figure legend corresponding to Fig. 6, do the authors mean ngrJ rather than gouJ?
3. What are the putative functions of NgrA and NgrT as predicted by BLAST? It may be beneficial for the authors to include a table of biosynthetic genes from the nigericin cluster and their putative functions in the SI.
Author Response
The manuscript submitted by Wei and coworkers details the investigation of a putative regulator (NgrR) of the nigericin biosynthetic gene cluster present in Streptomyces malaysiensis F913. The authors predict that the NgrR protein belongs to the SARP family of regulators using sequence analysis, and show that disruption of the ngrR gene abolishes nigericin production and transcription of nigericin biosynthetic genes. Overexpression of NgrR using kasO and hrdB promoters resulted in an increase in nigericin production relative to the wild-type strain.
General Comments
Engineering promoter systems in Streptomyces to boost the production of natural products has proven to be a successful strategy in many instances, and so the characterization of regulatory genes associated with BGCs of interest such as the nigericin BGC is clearly of value to the field. It would've been interesting to see if introducing multiple copies of ngrR into the strain would've further boosted the production (as has been shown in other examples), but perhaps this is beyond the scope of this study.
Response:This is a good suggestion. We expect that extra copies of ngrR will further improve nigericin production. This will be presented in a separate paper.
Specific Comments
- From the chromatograms shown in Fig. 3, there's another peak (RT 18 mins) which appears to increase in intensity when ngrRis disrupted. Do the authors know which metabolite this peak corresponds to? Is this peak also associated with the nigericin cluster? The authors should comment on this (perhaps as a note in the figure legend if it's not significant).
Response:Our collaborators are trying to purify enough amount of this compound to analysis its structure, currently their data indicated that this compound have no relationship with nigericin. We assumed that this peak may be another secondary metabolite which may compete for the same precursor with nigericin, overproducing of this compound contribute to enhanced supplement of precursor.
- In the figure legend corresponding to Fig. 6, do the authors mean ngrJrather than gouJ?
Response:We made the correction as suggested.
- What are the putative functions of NgrA and NgrT as predicted by BLAST? It may be beneficial for the authors to include a table of biosynthetic genes from the nigericin cluster and their putative functions in the SI.
Response: Based on suggestion of other reviewer, we have changed gene names to the same as proposed by the original discovered gene cluster in Streptomyces sp. DSM4137. NgrA and NgrT are now renamed as orf9 and orfL1, respectively. The function of Orf9 is unknown, while OrfL1 is a putative Prolyl-dipeptidyl aminopeptidase. The deduced functions of genes in the nigericin biosynthetic gene cluster have been reported in published reference (PMID: 17584617) and we have also added the information in the table S3.

Round 2
Reviewer 1 Report
The authors effectively improved the manuscript according to the review comments. However, this reviewer feels that several points need to be reconsidered.
1. Table S3 created in the revised version appears earlier than Tables S1 and S2. The numbering should be changed.
2. The authors added the comment regarding the occurrence of a consensus sequence in the promoter regions of the nig genes in the Discussion. This information should be described as part of Results, probably in the section in 2.3. It will be useful if the alignment of the consensus sequences together with their location in reference to the translational start position of the adjacent downstream coding sequence are shown as a new figure.
Author Response
The authors effectively improved the manuscript according to the review comments. However, this reviewer feels that several points need to be reconsidered.
- Table S3 created in the revised version appears earlier than Tables S1 and S2. The numbering should be changed.
Response: We have change the numbering of these tables as suggested.
- The authors added the comment regarding the occurrence of a consensus sequence in the promoter regions of the nig genes in the Discussion. This information should be described as part of Results, probably in the section in 2.3. It will be useful if the alignment of the consensus sequences together with their location in reference to the translational start position of the adjacent downstream coding sequence are shown as a new figure.
Response: We thank the reviewer’s good suggestion on our manuscript. We have moved the information into the section 2.3, and added a new figure (Figure S1) to show the consensus sequences together with their location in reference to the translational start position. As nigD-C and nigAI-nigE divergently transcribed, we showed both strands of the sequences, which makes it difficult to show alignment of different sequences, thus we showed the consensus sequences information separately in figure S1.